# POINT CLOUD GAN

**Chun-Liang Li**\*, **Manzil Zaheer**\*, **Yang Zhang, Barnabás Póczos, Ruslan Salakhutdinov**
Machine Learning Department, Carnegie Mellon University, Pittsburgh, PA 15213
`{chunlial,manzilz,yz6,bapoczos,rsalakhu}@cs.cmu.edu`

## ABSTRACT

Generative Adversarial Networks (GAN) can achieve promising performance on learning complex data distributions on different types of data. In this paper, we first show that a straightforward extension of an existing GAN algorithm is not applicable to point clouds, because the constraint required for discriminators is undefined for *set data*. We propose a two fold modification to a GAN algorithm to be able to generate point clouds (PC-GAN). First, we combine ideas from hierarchical Bayesian modeling and implicit generative models by learning a hierarchical and interpretable sampling process. A key component of our method is that we train a posterior inference network for the hidden variables. Second, PC-GAN defines a generic framework that can incorporate many existing GAN algorithms. We further propose a *sandwiching* objective, which results in a tighter Wasserstein distance estimate than the commonly used dual form in WGAN. We validate our claims on the ModelNet40 benchmark dataset and observe that PC-GAN trained by the sandwiching objective achieves better results on test data than existing methods. We also conduct studies on several tasks, including generalization on unseen point clouds, latent space interpolation, classification, and image to point clouds transformation, to demonstrate the versatility of the proposed PC-GAN algorithm.

## 1 INTRODUCTION

A fundamental problem in machine learning is that given a data set, learn a generative model that can efficiently generate *arbitrary many new sample points* from the domain of the underlying distribution (Bishop, 2006). Deep generative models use deep neural networks as a tool for learning complex data distributions (Kingma & Welling, 2013; Oord et al., 2016; Goodfellow et al., 2014). Especially, Generative Adversarial Networks (GAN) (Goodfellow et al., 2014) has drawn attention because of its success in many applications. Compelling results have been demonstrated on different types of data, including text, images, and videos (Lamb et al., 2016; Karras et al., 2017; Vondrick et al., 2016). Their wide range of applicability was also shown in many important problems, including data augmentation (Salimans et al., 2016), image style transformation (Zhu et al., 2017), image captioning (Dai et al., 2017), and art creations (Kang, 2017).

Recently, capturing 3D information is garnering attention. There are many different data types for 3D information, such as CAD, 3D meshes, and point clouds. 3D point clouds are getting popular since these store more information than 2D images and sensors capable of collecting point clouds have become more accessible. These include Lidar on self-driving cars, Kinect for Xbox, and face identification sensor on phones. Compared to other formats, point clouds can be easily represented as a set of points, which has several advantages, such as permutation invariance of the set members. The algorithms which can effectively learn from this type of data is an emerging field (Qi et al., 2017a;b; Zaheer et al., 2017; Kalogerakis et al., 2017; Fan et al., 2017). However, compared to supervised learning, unsupervised generative models for 3D data are still under explored (Achlioptas et al., 2017; Oliva et al., 2018).

Extending existing GAN frameworks to point clouds or more generally set data is not straightforward. In this paper, we begin by formally defining the problem and discussing its difficulty (Section 2). Circumventing the challenges, we propose a deep generative adversarial network (PC-GAN) with a hierarchical sampling and inference network for point clouds. The proposed architecture learns a stochastic procedure which can generate new point clouds and draw samples from the generated point clouds without explicitly modeling the underlying density function (Section 3). The proposed

PC-GAN is a generic algorithm which can incorporate many existing GAN variants. By utilizing the property of point clouds, we further propose a *sandwiching* objective by considering both upper and lower bounds of Wasserstein distance estimate, which can lead to tighter approximation (Section 3.1). Evaluation on ModelNet40 shows excellent generalization capability of PC-GAN. We first demonstrate that we can sample from the learned model to generate new point clouds and the latent representations learned by the inference network provide meaningful interpolations between point clouds. Then we show the conditional generation results on *unseen* classes of objects, which demonstrates the superior generalization ability of PC-GAN. Lastly, we also provide several interesting studies, such as classification and point clouds generation from images (Section 5).

## 2 PROBLEM DEFINITION AND DIFFICULTY

A point cloud for an object $\theta$ is a *set* of $n$ low dimensional vectors $X = \{x_1, ..., x_n\}$ with $x_i \in \mathbb{R}^d$, where $d$ is usually 3 and $n$ can be infinite. $M$ different objects can be described as a collection of point clouds $X^{(1)}, ..., X^{(M)}$. A generative model for sets should be able to: (1) Sample entirely new sets according to $p(X)$, and (2) sample arbitrarily many more points from the distribution of given set, i.e. $x \sim p(x|X)$.

Based on the De-Finetti theorem, we could factor the probability with some suitably defined $\theta$, such as object representation of point clouds, as $p(X) = \int_\theta \prod_{i=1}^n p(x_i|\theta)p(\theta)d\theta$. In this view, the factoring can be understood as follows: Given an object, $\theta$, the points $x_i$ in the point cloud can be considered as i.i.d. samples from $p(x|\theta)$, an unknown latent distribution representing object $\theta$. Joint likelihood can be expressed as:

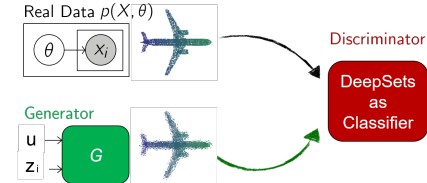

$$p(X, \theta) = \underbrace{p(\theta)}_{\text{object}} \underbrace{\prod_{i=1}^n p(x_i|\theta)}_{\text{points for object}} \quad (1)$$

Figure 1: Natural extension of GAN to handle set data does not work.

One approach can be used to model the distribution of the point cloud set together, i.e., $\{\{x_i^{(1)}\}_{i=1}^n, ..., \{x_i^{(m)}\}_{i=1}^n\}$. In this setting, a naíve application of traditional GAN is possible through treating the point cloud as finite dimensional vector by fixing the number and order of the points (reducing the problem to instances in $\mathbb{R}^{n \times 3}$) with DeepSets (Zaheer et al., 2017) classifier as the discriminator to distinguish real sets from fake sets. However, this approach would not work in practice because the *integral probability metric* (IPM) guarantees behind the traditional GAN no longer hold (e.g. in case of Arjovsky et al. (2017), nor are 1-Lipschitz functions over sets well-defined). The probabilistic divergence approximated by a DeepSets classifier might be ill-defined. Counter examples for breaking IPM guarantees can be easily found as we show next.

**Counter Example** Consider a simple GAN (Goodfellow et al., 2014) with a DeepSets classifier as the discriminator. In order to generate coherent sets of variable size, we consider a generator $G$ having two noise sources: $u$ and $z_i$. To generate a set, $u$ is sampled once and $z_i$ is sampled for $i = 1, 2, ..., n$ to produce $n$ points in the generated set. Intuitively, fixing the first noise source $u$ selects a set and ensures the points generated by repeated sampling of $z_i$ are coherent and belong to the same set. The setup is depicted in Figure 1. In this setup, the GAN minimax problem would be:

$$\min_G \max_D \mathop{\mathbb{E}}_{\substack{\theta \sim p(\theta) \\ x_i \sim p(x_i|\theta)}} \left[ \log D\left( \{x_i\} \right) \right] + \mathop{\mathbb{E}}_{\substack{u \sim p(u) \\ z_i \sim p(z_i)}} \left[ \log \left( 1 - D\left( \{G(u, z_i)\} \right) \right) \right] \quad (2)$$

Now consider the case, when there exists an 'oracle' mapping $T$ which maps each sample point deterministically to the object it originated from, i.e. $\exists T : T(\{x_i\}) = \theta$. A valid example is when different $\theta$ leads to conditional distribution $p(x|\theta)$ with non-overlapping support. Let $D = D' \circ T$ and $G$ ignore $z$, then the optimization task becomes as follows:

$$\min_G \max_{D'} \mathop{\mathbb{E}}_{\substack{\theta \sim p(\theta) \\ x_i \sim p(x_i|\theta)}} \left[ \log D'\left( T\left( \{x_i\} \right) \right) \right] + \mathop{\mathbb{E}}_{\substack{u \sim p(u) \\ z_i \sim p(z_i)}} \left[ \log \left( 1 - D'\left( T\left( \{G(u, z_i)\} \right) \right) \right) \right]$$

$$\Rightarrow \min_G \max_{D'} \mathop{\mathbb{E}}_{\substack{\theta \sim p(\theta) \\ x_i \sim p(x_i|\theta)}} \left[ \log D'\left( \theta \right) \right] + \mathop{\mathbb{E}}_{\substack{u \sim p(u) \\ z_i \sim p(z_i)}} \left[ \log \left( 1 - D'\left( T\left( \{G(u)\} \right) \right) \right) \right] \quad (3)$$

$$\Rightarrow \min_G \max_{D'} \mathop{\mathbb{E}}_{\theta \sim p(\theta)} \left[ \log D'\left( \theta \right) \right] + \mathop{\mathbb{E}}_{u \sim p(u)} \left[ \log \left( 1 - D'\left( T\left( \{G(u)\} \right) \right) \right) \right]$$

Thus, we can achieve the lower bound $-\log(4)$ by only matching the $p(\theta)$ component, while the conditional $p(x|\theta)$ is allowed to remain arbitrary. So simply using DeepSets classifier *without any constraints* in simple GAN in order to handle sets does not lead to a valid generative model.

## 3  PROPOSED METHOD

As described in Section 2, directly learning point cloud generation under GAN formulation is difficult. However, given $\theta$, learning $p(x|\theta)$ is a simpler task of learning a 3-dimensional distribution. Given two point clouds, one popular heuristic distance between them is the Chamfer distance (Achlioptas et al., 2017). On the other hand, if we treat each point cloud as a *3-dimensional distribution*, we can adopt a broader class of probabilistic divergences for comparing them. Instead of learning explicit densities (Jian & Vemuri, 2005; Strom et al., 2010; Eckart et al., 2015), we are interested in implicit generative models with a GAN-like objective (Goodfellow et al., 2014), which has

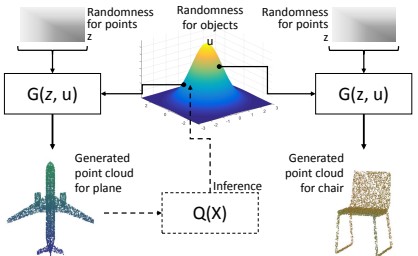

Figure 2: Overview of PC-GAN.

been demonstrated to learn complicated distributions. Formally, given a $\theta$, we train a generator $G_x(z, \theta)$ such that $x = G_x(z, \theta)$, where $z \sim p(z)$. The generator $G_x(z, \theta)$ follows $\mathbb{G}$ by optimizing a probabilistic divergence $D(\mathbb{P}\|\mathbb{G})$ between the distribution $\mathbb{G}$ of $G_x(z, \theta)$ and $p(x|\theta)$, which is denoted as $\mathbb{P}$. The full objective can be written as $\mathbb{E}_{\theta \sim p(\theta)} \left[ \min_{G_x} D(\mathbb{P}\|\mathbb{G}) \right]$.

**Inference**   Although GANs have been extended to learn conditional distributions (Mirza & Osindero, 2014; Isola et al., 2017), they require conditioning variables to be observed, such as the one-hot label or a given image. Our $\theta$, instead, is an unobserved latent variable for modeling different objects, which we need to infer during training. The proposed algorithm has to concurrently learn the inference network $Q(X) \approx \theta$ while we learn $p(x|\theta)$. Since $X$ is a set of points, we can adopt Qi et al. (2017a); Zaheer et al. (2017) for modeling $Q$. We provide more discussion on this topic in the Appendix A.1.

**Hierarchical Sampling**   After training $G_x$ and $Q$, we use the trained $Q$ to collect the inferred $Q(X)$ and train the generator $G_\theta(u) \sim p(\theta)$ for higher hierarchical sampling. Here $u \sim p(u)$ is the other noise source independent of $z$. In addition to layer-wise training, a joint training could further boost performance. The full generative process for sampling one point cloud could be represented as

$$\{x_i\}_{i=1}^n = \{G(z_i, u)\}_{i=1}^n = \{G_x(z_i, G_\theta(u))\}_{i=1}^n, \text{ where } z_1, \ldots, z_n \sim p(z), \text{ and } u \sim p(u).$$

The overview of proposed algorithm for point cloud generation (*PC-GAN*) is shown in Figure 2.

### 3.1  DIFFERENT DIVERGENCES FOR MATCHING POINT CLOUDS

To train the generator $G_x$ using a GAN-like objective for point clouds, we need a discriminator $f(\cdot)$ to distinguishes generated samples and true samples conditioned on $\theta$. Combining with the inference network $Q(X)$ discussed aforementioned, the objecitve with IPM-based GANs can be written as

$$\mathbb{E}_{\theta \sim p(\theta)} \left[ \min_{G_x, Q} \max_{f \in \Omega_f} \underbrace{\mathbb{E}_{x \sim p(X|\theta)} [f(x)] - \mathbb{E}_{z \sim p(z), X \sim p(X|\theta)} [f(G_x(z, Q(X)))]}_{D(\mathbb{P}\|\mathbb{G})} \right], \qquad (4)$$

where $\Omega_f$ is the constraint for different probabilistic distances, such as 1-Lipschitz (Arjovsky et al., 2017), $L^2$ ball (Mroueh & Sercu, 2017) or Sobolev ball (Mroueh et al., 2017).

### 3.2  TIGHTER SOLUTIONS VIA SANDWICHING

In our setting, each point $x_i$ in the point cloud can be considered to correspond to single images when we train GANs over images. An example is illustrated in Figure 3 where samples from MMD-GAN (Li et al., 2017a) trained on CelebA consists of both good and bad faces. In case of images, when quality is evaluated, it primarily focuses on coherence individual images and the few bad ones are usually left out. Whereas in case of point cloud, to get representation of an object we need many sampled points together and presence of outlier points degrades the quality of the object. Thus, when training a generative model for point cloud, we need to ensure a much lower distance $D(\mathbb{P}\|\mathbb{G})$ between true distribution $\mathbb{P}$ and generator distribution $\mathbb{G}$ than would be needed in case of images.

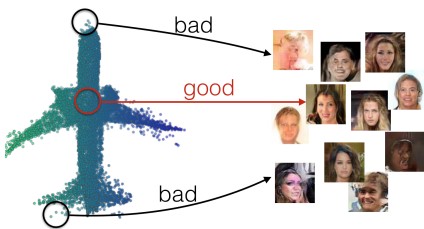

We begin by noting that the popular Wasserstein GAN (Arjovsky et al., 2017), aims to optimize $G$ by $\min w(\mathbb{P}, \mathbb{G})$, where $w(\mathbb{P}, \mathbb{G})$ is the Wasserstein distance $w(\mathbb{P}, \mathbb{G})$ between the truth $\mathbb{P}$ and generated distribution $\mathbb{G}$ of $G$. Many GAN works (e.g. Arjovsky et al. (2017)) approximate $w(\mathbb{P}, \mathbb{G})$ in dual form (a maximization problem), such as (4), by neural networks. The resulting estimate $W_L(\mathbb{P}, \mathbb{G})$ is a lower bound of the true Wasserstein distance, as neural networks can only recover a subset of 1-Lipschitz functions (Arora et al., 2017) required in the

Figure 3: Connection between good/bad points and faces generated from a GAN.

dual form. However, finding a lower bound $W_L(\mathbb{P}, \mathbb{G})$ for $w(\mathbb{P}, \mathbb{G})$ may not be an ideal surrogate for solving a minimization problem $\min w(\mathbb{P}, \mathbb{G})$. In optimal transport literature, Wassertein distance is usually estimated by approximate matching cost, $W_U(\mathbb{P}, \mathbb{G})$, which gives us an upper bound of the true Wasserstein distance.

We propose to combine, in general, a lower bound $W_L$ and upper bound estimate $W_U$ by sandwiching the solution between the two, i.e. we solve the following minimization problem:

$$\min_{G} \quad W_U(\mathbb{P}, \mathbb{G}) \qquad \text{s.t.} \quad W_U(\mathbb{P}, \mathbb{G}) - W_L(\mathbb{P}, \mathbb{G}) < \lambda \tag{5}$$

The problem can be simplified and solved using method of lagrange multipliers as follows:

$$\min_{G} W_s(\mathbb{P}, \mathbb{G}) := (1 - s)W_U(\mathbb{P}, \mathbb{G}) + sW_L(\mathbb{P}, \mathbb{G}) \tag{6}$$

By solving the new *sandwiched problem* (6), we show that under certain conditions we obtain a better estimate of Wasserstein distance in the following lemma:

**Lemma 1.** *Suppose we have two approximators to Wasserstein distance: an upper bound $W_U$ and a lower $W_L$, such that $\forall \mathbb{P}, \mathbb{G} : (1 + \epsilon_1)w(\mathbb{P}, \mathbb{G}) \leq W_U(\mathbb{P}, \mathbb{G}) \leq (1 + \epsilon_2)w(\mathbb{P}, \mathbb{G})$ and $\forall P, G : (1 - \epsilon_2)w(\mathbb{P}, \mathbb{G}) \leq W_L(\mathbb{P}, \mathbb{G}) \leq (1 - \epsilon_1)w(\mathbb{P}, \mathbb{G})$ respectively, for some $\epsilon_2 > \epsilon_1 > 0$ and $\epsilon_1 > \epsilon_2/3$. Then, using the sandwiched estimator $W_s$ from (6), we can achieve tighter estimate of the Wasserstein distance than using either one estimator, i.e.*

$$\exists s : |W_s(\mathbb{P}, \mathbb{G}) - w(\mathbb{P}, \mathbb{G})| < \min\{|W_U(\mathbb{P}, \mathbb{G}) - w(\mathbb{P}, \mathbb{G})|, |W_L(\mathbb{P}, \mathbb{G}) - w(\mathbb{P}, \mathbb{G})|\} \tag{7}$$

### 3.2.1 UPPER AND LOWER BOUND IMPLEMENTATION

For $W_L$, we can adopt many GAN variants (Arjovsky et al., 2017; Gulrajani et al., 2017; Mroueh & Sercu, 2017). For $W_U$, we use Bertsekas (1985), which results in a fast $\epsilon$ approximation of the Wasserstein distance estimate in primal form without solving non-trivial linear programming. We remark estimating Wasserstein distance $w(\mathbb{P}, \mathbb{G})$ with *finite samples* via its primal is only favorable to low dimensional data, such as point clouds. The error of empirical estimate in primal is $O(1/n^{1/d})$ (Weed & Bach, 2017). When the dimension $d$ is large (e.g. images), we cannot accurately estimate $w(\mathbb{P}, \mathbb{G})$ in primal as well as its upper bound with a small minibatch. For detailed discussion of finding lower and upper bound, please refer to Appendix A.2 and A.3.

## 4 RELATED WORKS

Generative Adversarial Network (Goodfellow et al., 2014) aims to learn a generator that can sample data followed by the data distribution. Compelling results on learning complex data distributions with GAN have been shown on images (Karras et al., 2017), speech (Lamb et al., 2016), text (Yu et al., 2016; Hjelm et al., 2017), vedio (Vondrick et al., 2016) and 3D voxels (Wu et al., 2016). However, the GAN algorithm on 3D point cloud is still under explored (Achlioptas et al., 2017). Many alternative objectives for training GANs have been studied. Most of them are the *dual form* of $f$-divergence (Goodfellow et al., 2014; Mao et al., 2017; Nowozin et al., 2016), integral probability metrics (IPMs) (Zhao et al., 2016; Li et al., 2017a; Arjovsky et al., 2017; Gulrajani et al., 2017) or IPM extensions (Mroueh & Sercu, 2017; Mroueh et al., 2017). Genevay et al. (2018) learn the generative model by the approximated primal form of Wasserstein distance (Cuturi, 2013).

Instead of training a generative model on the data space directly, one popular approach is combining with autoencoder (AE), which is called adversarial autoencoder (AAE) (Makhzani et al., 2015). AAE constrain the encoded data to follow normal distribution via GAN loss, which is similar to VAE (Kingma & Welling, 2013) by replacing the KL-divergence on latent space via any GAN loss.

Tolstikhin et al. (2017) provide a theoretical explanation for AAE by connecting it with the primal form of Wasserstein distance. The other variant of AAE is training the other generative model to learn the distribution of the encoded data instead of enforcing it to be similar to a known distribution (Engel et al., 2017; Kim et al., 2017). Achlioptas et al. (2017) explore a AAE variant for point cloud. They use a specially-designed encoder network (Qi et al., 2017a) for learning a compressed representation for point clouds before training GAN on the latent space. However, their decoder is restricted to be a MLP which generates $m$ fixed number of points, where $m$ has to be pre-defined. That is, the output of their decoder is fixed to be $3m$ for 3D point clouds, while the output of the proposed $G_x$ is only 3 dimensional and $G_x$ can generate arbitrarily many points by sampling different random noise $z$ as input. Yang et al. (2018); Groueix et al. (2018b) propose similar decoders to $G_x$ with fixed grids to break the limitation of Achlioptas et al. (2017) aforementioned, but they use heuristic Chamfer distance without any theoretical guarantee and do not exploit generative models for point clouds. The proposed PC-GAN can also be interpreted as an encoder-decoder formulation. However, the underlying interpretation is different. We start from De-Finetti theorem to learn both $p(X|\theta)$ and $p(\theta)$ with inference network interpretation of $Q$, while Achlioptas et al. (2017) focus on learning $p(\theta)$ without modeling $p(X|\theta)$.

Lastly, GAN for learning conditional distribution (conditional GAN) has been studied in images with single conditioning (Mirza & Osindero, 2014; Pathak et al., 2016; Isola et al., 2017; Chang et al., 2017) or multiple conditioning (Wang & Gupta, 2016). The case on point cloud is still under explored. Also, most of the works assume the conditioning is given (e.g. labels and base images) without learning the inference during the training. Training GAN with inference is studied by Donahue et al. (2016); Dumoulin et al. (2016); Li et al. (2017b); however, their goal is to infer the random noise $z$ of generators and match the semantic latent variable to be similar to $z$. Li et al. (2018) is a parallel work aiming to learn GAN and unseen latent variable simultaneously, but they only study image and video datasets.

## 5 EXPERIMENTS

In this section we demonstrate the point cloud generation capabilities of PC-GAN. As discussed in Section 4, we refer Achlioptas et al. (2017) as AAE as it could be treated as an AAE extension to point clouds and we use the implementation provided by the authors for experiments. The sandwitching objective $W_s$ for PC-GAN combines $W_L$ and $W_U$ with the mixture 1:20 without tunning for all experiment. $W_L$ is a GAN loss by combining Arjovsky et al. (2017) and Mroueh & Sercu (2017) (technical details are in Appendix A.3) and we adopt (Bertsekas, 1985) for $W_U$. We parametrize $Q$ in PC-GAN by DeepSets (Zaheer et al., 2017). The review of DeepSets is in Appendix E. Other detailed configurations of each experiment can be found in Appendix F.

### 5.1 SYNTHETIC DATASETS

We generate 2D circle point clouds. The center of circles follows a mixture of Gaussians $\mathcal{N}(\{\pm16\} \times \{\pm16\}, 16I)$ with equal mixture weights. The radius of the circles was drawn from a uniform distribution $Unif(1.6, 6.4)$. One sampled circile is shown in Figure 4a.

For AAE, the output size of the decoder is $500 \times 2$ for 500 points, and the output size of the encoder (latent code) is 20. The total number of parameters are $24K$. For PC-GAN, the inference network output size is 15. The total nuumber of parameters of PC-GAN is only $12K$. We evaluated the conditional distributions on the $10,000$ testing circles. We measured the empirical distributions of the centers and the radius of the generated circles conditioning on the testing data as shown in Figure 4.

From Figure 4, both AAE and PC-GAN can successfully recover the center distribution, but AAE does not learn the radius distribution well even with larger latent code

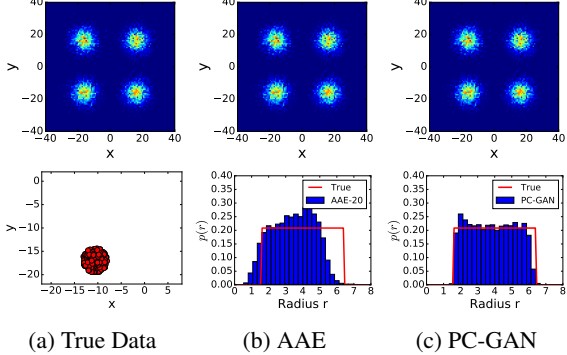

(a) True Data      (b) AAE      (c) PC-GAN

Figure 4: (a) (top) the true center distribution and (bottom) one example of a circle point cloud. (b-d) are the reconstructed center and radius distributions.

Table 1: Quantitative results of different models trained on different subsets of ModelNet40 and evaluated on the corresponding test set. ModelNet10 is a subset containing 10 classes of objects, while ModelNet40 is a full training set. AAE is trained using the code from Achlioptas et al. (2017). The PC-GAN variants are trained via upper bound $W_U$, lower bound $W_L$ and sandwiching loss $W_s$.

| Data | Distance to Face (D2F $\downarrow$) | | | | Coverage ($\uparrow$) | | | |
|---|---|---|---|---|---|---|---|---|
| | PC-GAN ($W_s$) | AAE | PC-GAN ($W_U$) | PC-GAN ($W_L$) | PC-GAN ($W_s$) | AAE | PC-GAN ($W_U$) | PC-GAN ($W_L$) |
| Aeroplanes | 1.89E+01 | 1.99E+01 | 1.53E+01 | 2.49E+01 | 1.95E-01 | 2.99E-02 | 1.73E-01 | 1.88E-01 |
| Benches | 1.09E+01 | 1.41E+01 | 1.05E+01 | 2.46E+01 | 4.44E-01 | 2.35E-01 | 2.58E-01 | 3.83E-01 |
| Cars | 4.39E+01 | 6.23E+01 | 4.25E+01 | 6.68E+01 | 2.35E-01 | 4.98E-02 | 1.78E-01 | 2.35E-01 |
| Chairs | 1.01E+01 | 1.08E+01 | 1.06E+01 | 1.08E+01 | 3.90E-01 | 1.82E-01 | 3.57E-01 | 3.95E-01 |
| Cups | 1.44E+03 | 1.79E+03 | 1.28E+03 | 3.01E+03 | 6.31E-01 | 3.31E-01 | 4.32E-01 | 5.68E-01 |
| Guitars | 2.16E+02 | 1.93E+02 | 1.97E+02 | 1.81E+02 | 2.25E-01 | 7.98E-02 | 2.11E-01 | 2.27E-01 |
| Lamps | 1.47E+03 | 1.60E+03 | 1.64E+03 | 2.77E+03 | 3.89E-01 | 2.33E-01 | 3.79E-01 | 3.66E-01 |
| Laptops | 2.43E+00 | 3.73E+00 | 2.65E+00 | 2.58E+00 | 4.31E-01 | 2.56E-01 | 3.93E-01 | 4.55E-01 |
| Sofa | 1.71E+01 | 1.64E+01 | 1.45E+01 | 2.76E+01 | 3.65E-01 | 1.62E-01 | 2.94E-01 | 3.47E-01 |
| Tables | 2.79E+00 | 2.96E+00 | 2.44E+00 | 3.69E+00 | 3.82E-01 | 2.59E-01 | 3.20E-01 | 3.53E-01 |
| ModelNet10 | 5.77E+00 | 6.89E+00 | 6.03E+00 | 9.19E+00 | 3.47E-01 | 1.90E-01 | 3.36E-01 | 3.67E-01 |
| ModelNet40 | 4.84E+01 | 5.86E+01 | 5.24E+01 | 7.96E+01 | 3.80E-01 | 1.85E-01 | 3.65E-01 | 3.71E-01 |

(20) and more parameters ($24K$). The gap of memory usage could be larger if we configure AAE to generate more points, while the model size required for PC-GAN is independent of the number of points. The reason is MLP decoder adopted by Achlioptas et al. (2017) wastes parameters for nearby points. Using the much larger model (more parameters) could boost the performance. However, it is still restricted to generate a fixed number of points for each object as we discussed in Section 4.

## 5.2 STUDY ON MODELNET40

We consider ModelNet40 (Wu et al., 2015) benchmark, which contains 40 classes of objects. There are $9,843$ training and $2,468$ testing instances. We follow Achlioptas et al. (2017) to consider two settings. One is training on single class of objects. The other is training on all $9,843$ objects in the training set. Achlioptas et al. (2017) set the latent code size of AAE to be $128$ and $256$ for these two settings, with the total number of parameters to be $15M$ and $15.2M$, respectively. Similarly, we set the output dimension of $Q$ in PC-GAN to be $128$ and $256$ for single-class and all-classes. The total number of parameters are $1M$ and $3M$, respectively.

**Metrics for Quantitative Comparison** Firstly, we are interested in whether the learned $G_x$ and $Q$ can model the distribution of unseen test data. For each test point cloud, we infer the latent variable $Q(X)$, then use $G_x$ to generate points. We then compare the distribution between the input point cloud and the conditionally generated point clouds.

There are many finite sample estimation for $f$-divergence and IPM can be used for evaluation. However, those estimators with finite samples are either biased or with high variance (Peyré et al., 2017; Wang et al., 2009; Póczos et al., 2012; Weed & Bach, 2017). Also, it is impossible to use these estimators with infinitely many samples if they are accessible.

For ModelNet40, the meshes of each object are available. In many statistically guaranteed distance estimates, the adopted statistics are commonly based on distance between nearest neighbors (Wang et al., 2009; Póczos et al., 2012). Therefore, we propose to measure the performance with the following criteria. Given a point cloud $\{x_i\}_{i=1}^n$ and a mesh, which is a collection of faces $\{F_j\}_{j=1}^m$, we measure the *distance to face (D2F)* as

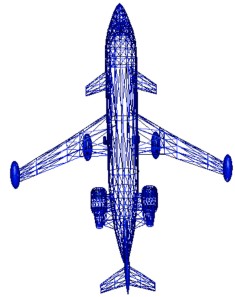

Figure 5: Sample mesh of ModelNet40

$$D2F\left(\{x_i\}_{i=1}^n, \{F_j\}_{j=1}^m\right) = \frac{1}{n}\sum_{i=1}^n \min_j \mathcal{D}(x_i, F_j),$$

where $\mathcal{D}(x_i, F_j)$ is the Euclidean distance from $x_i$ to the face $F_j$. This distance is similar to Chamfer distance, which is commonly used for measuring images and point clouds (Achlioptas et al., 2017; Fan et al., 2017), with infinitely samples from true distributions (meshes).

Nevertheless, the algorithm can have low or zero D2F by only focusing a small portion of the point clouds (mode collapse). Therefore, we are also interested in whether the generated points recover enough supports of the distribution. We compute the *Coverage* ratio as follows. For each point, we

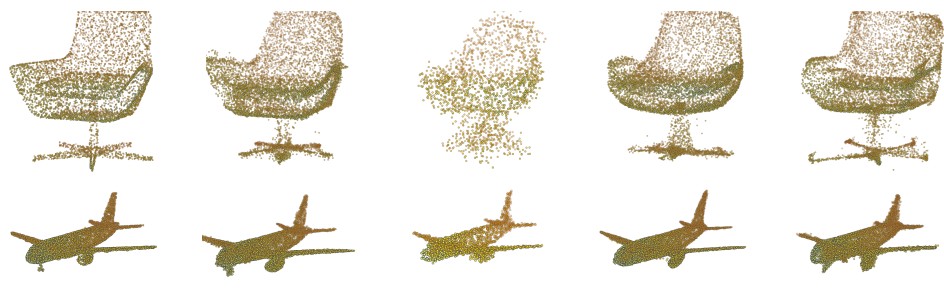

| (a) Data | (b) PC-GAN ($W_s$) | (c) AAE | (d) PC-GAN ($W_U$) | (e) PC-GAN ($W_L$) |

Figure 6: Example reconstruction (conditional generation) on test objects. PC-GAN with sandwiching ($W_s$) is better in capturing fine details like wheels of aeroplane or proper chair legs.

find the its nearest face, we then treat this face is covered[1]. We then compute the ratio of number of faces of a mesh is covered. A sampled mesh is showed in Figure 5, where the details have more faces (non-uniform). Thus, it is difficult to get high coverage for AAE or PC-GAN trained by limited number of sampled points. However, the coverage ratio, on the other hand, serve as an indicator about how much details the model recovers.

The results are reported in Table 1. We compare four different algorithm, AAE and PC-GAN with three objectives, including upper bound $W_U$ ( $\epsilon$ approximated Wasserstein distance), lower bound $W_L$ (GAN with $L^2$ ball constraints and weight clipping), and the sandwiching loss $W_s$ as discussed in Section 3.2, The study with $W_U$ and $W_L$ also serves as the ablation test of $W_s$.

**Comparison between Upper bound, Lower bound and Sandwiching** Since $W_U$ directly optimizes distance between training and generated point clouds, $W_U$ usually results in smaller D2F than $W_L$ in Table 1. One the other hand, although $W_L$ only recovers lower bound estimate of Wasserstein distance, its discriminator is known to focus on learning support of the distribution (Bengio, 2018), which results in better coverage (support) than $W_U$.

Theoretically, the proposed sandwiching $W_s$ results in a tighter Wasserstein distance estimation than $W_U$ and $W_L$ (Lemma 1). Based on above discussion, it can also be understood as balancing both D2F and coverage by combining both $W_U$ and $W_L$ to get a desirable middle ground. Empirically, we even observe that $W_s$ results in better coverage than $W_L$, and competitive D2F with $W_U$. The intuitive explanation is that some discriminative tasks are *off* to $W_U$ objective, so the discriminator can focus more on learning distribution supports. We argue that this difference is crucial for capturing the object details. Some reconstructed point clouds of testing data are shown in Figure 6. For aeroplane examples, $W_U$ are failed to capture aeroplane tires and $W_s$ has better tire than $W_L$. For Chair example, $W_s$ recovers better legs than $W_U$ and better seat cushion than $W_L$. Lastly, we highlight $W_s$ outperforms others more significantly when training data is larger (ModelNet10 and ModelNet40) in Table 1.

**Comparison between PC-GAN and AAE** In most of cases, PC-GAN with $W_s$ has lower D2F in Table 1 with less number of parameters aforementioned. Similar to the argument in Section 5.1, although AAE use larger networks, the decoder wastes parameters for nearby points. AAE only outperforms PC-GAN ($W_s$) in Guitar and Sofa in terms of D2F, since the variety of these two classes are low. It is easier for MLP to learn the shared template (basis) of the point clouds. On the other hand, due to the limitation of the fixed number of output points and Chamfer distance objective, AAE has worse coverage than PC-GAN, It can be supported by Figure 6, where AAE is also failed to recover aeroplane tire.

**Hierarchical Sampling** In Section 3, we propose a hierarchical sampling process for sampling point clouds. In the first hierarchy, the generator $G_\theta$, samples a object ($\theta = G_\theta(u), u \sim \mathbb{P}(u)$), while the second generator $G_x$ samples points based on $\theta$ to form the point cloud.

The randomly sampled results without given any data as input are shown in Figure 7. More results can be found in Appendix C. The point clouds are all smooth, structured and almost symmetric. It shows PC-GAN captures inherent symmetries and patterns in all the randomly sampled objects, even if overall object is not perfectly formed. This highlights that learning point-wise generation scheme encourages learning basic building blocks of objects.

---

[1] We should do thresholding to ignore outlier points. In our experiments, we observe that without excluding outliers does not change conclusion for comparison.

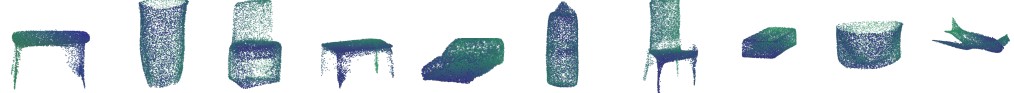

Figure 7: Randomly sampled objects and corresponding point cloud from the hierarchical sampling Even if there are some defects, the objects are smooth, symmetric and structured.

**Interpolation of Learned Manifold**    We study whether the interpolation between two objects on the latent space results in smooth change. We interpolate the inferred representations of two objects by $Q$, and use the generator $G_x$ to sample points based on the interpolation. The inter-class result is shown in Figure 8. More studies about interpolation between rotations can be found in Appendix D.1.

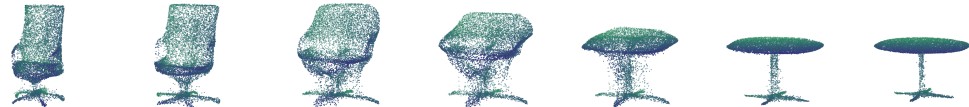

Figure 8: Interpolating between latent representations $Q(X)$ of a table and a chair point clouds.

**Generalization on Unseen Classes**    In above, we studied the reconstruction of unseen testing objects, while PC-GAN still saw the point clouds from the same class during training. *Here we study the more challenging task.* We train PC-GAN on first 30 (Alphabetic order) class, and test on the other *fully unseen 10 classes*. Some reconstructed (conditionally generated) point clouds are shown in Figure 9. More (larger) results can be found in Appendix C. For the object from the unseen classes, the conditionally generated point clouds still recovers main shape and reasonable geometry structure, which confirms the advantage of the proposed PC-GAN: by enforcing the point-wise transformation, the model is forced to learn the underlying geometry structure and the shared building blocks, instead of naively copying the input from the conditioning. The rsulted D2F and coverage are $57.4$ and $0.36$, which are only slightly worse than $48.4$ and $0.38$ by training on whole 40 classes in Table 1 (ModelNet40), which also supports the claims of the good generalization ability of PC-GAN.

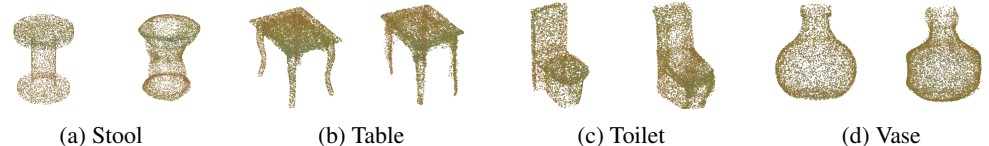

| (a) Stool | (b) Table | (c) Toilet | (d) Vase |

Figure 9: The reconstructed objects from unseen classes (even in training). In each plot, LHS is true data while RHS is PC-GAN. PC-GAN generalizes well as it can match patterns and symmetries from classes seen in the past to new unseen classes.

**More Studies**    We also condct other studies to make experiments complete, including interpolation between different rotations, classification and image to point clouds. Due to space limit, all of the results can be found in Appendix D.

## 6    CONCLUSION

In this paper, we first showed a straightforward extension of existing GAN algorithm is not applicable to point clouds. We then proposed a GAN modification (PC-GAN) that is capable of learning to generate point clouds by using ideas both from hierarchical Bayesian modeling and implicit generative models. We further propose a *sandwiching* objective which results in a tighter Wasserstein distance estimate theoretically and better performance empirically.

In contrast to some existing methods (Achlioptas et al., 2017), PC-GAN can generate arbitrary as many *i.i.d.* points as we need to form a point clouds without pre-specification. Quantitatively, PC-GAN achieves competitive or better results using smaller network than existing methods. We also demonstrated that PC-GAN can capture delicate details of point clouds and generalize well even on unseen data. Our method learns "point-wise" transformations which encourage the model to learn the building components of the objects, instead of just naively copying the whole object. We also demonstrate other interesting results, including point cloud interpolation and image to point clouds.

Although we only focused on 3D applications in this paper, our framework can be naturally generalized to higher dimensions. In the future we would like to explore higher dimensional applications, where each 3D point can have other attributes, such as RGB colors and 3D velocity vectors.

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

# A DETAILS OF THE PROPOSED METHOD

## A.1 NEURAL NETWORK REALIZATION OF INFERENCE NETWORK

Our solution comprises of a generator $G_x(z, \psi)$ which takes in a noise source $z \in \mathbb{R}^{d_1}$ and a descriptor $\psi \in \mathbb{R}^{d_2}$ encoding information about distribution of $\theta$. For a given $\theta_0$, the descriptor $\psi$ would encode information about the distribution $\delta(\theta - \theta_0)$ and samples generated as $x = G_x(z, \psi)$ would follow the distribution $p(x|\theta_0)$. More generally, $\psi$ can be used to encode more complicated distributions regarding $\theta$ as well. In particular, it could be used to encode the posterior $p(\theta|X)$ for a given sample set $X$, such that $x = G_x(z, \psi)$ follows the posterior predictive distribution:

$$p(x|X) = \int p(x|\theta)p(\theta|X)d\theta.$$

A major hurdle in taking this path is that $X$ is a set of points, which can vary in size and permutation of elements. Thus, making design of $Q$ complicated as traditional neural network can not handle this and possibly is the reason for absence of such framework in the literature despite being a natural solution for the important problem of generative modeling of point clouds. However, we can overcome this challenge and we propose to construct the inference network by utilizing the permutation equivariant layers from Deep Sets (Zaheer et al., 2017). This allows it handle variable number of inputs points in arbitrary order, yet yielding a consistent descriptor $\psi$.

After training $G_x$ and the inference network $Q$, we use trained $Q$ to collect inferred $Q(X)$ and train the generator $G_\theta(u) \sim p(\theta)$ for higher hierarchical sampling, where $u$ is the other noise source independent of $z$. In addition to the layer-wise training, a joint training may further boost the performance. The full generative process for sampling one point cloud could be represented as $\{x_i\}_{i=1}^n = \{G_x(z_i, G_\theta(u))\}_{i=1}^n$, where $z_1, \ldots, z_n \sim p(z)$ and $u \sim p(u)$.

We call the proposed GAN framework for learning to generative point clouds as *PC-GAN* as shown in Figure 2. The conditional distribution matching with a learned inference in PC-GAN can also be interpreted as an encoder-decoder formulation (Kingma & Welling, 2013). The difference between it and the point cloud autoencoder (Achlioptas et al., 2017; Yang et al., 2018) will be discussed in Section 4.

## A.2 UPPER IMPLEMENTATION

The primal form of Wasserstein distance is defined as

$$w(\mathbb{P}, \mathbb{G}) = \inf_{\gamma \in \Gamma(\mathbb{P}, \mathbb{G})} \int \|x - y\|_1 d\gamma(x, y),$$

where $\gamma$ is the *coupling* of $P$ and $G$. The Wasserstein distance is also known as optimal transport (OT) or earth moving distance (EMD). As the name suggests, when $w(\mathbb{P}, \mathbb{G})$ is estimated with finite number of samples $X = x_1, \ldots, x_n$ and $Y = y_1, \ldots, y_n$, we find the one-to-one matching between $X$ and $Y$ such that the total pairwise distance is minimal. The resulting minimal total (average) pairwise distance is $w(X, Y)$. In practice, finding the exact matching efficiently is non-trivial and still an open research problem (Peyré et al., 2017). Instead, we consider an approximation provided by Bertsekas (1985). It is an iterative algorithm where each iteration operates like an auction whereby unassigned points $x \in X$ bid simultaneously for closest points $y \in Y$, thereby raising their prices. Once all bids are in, points are awarded to the highest bidder. The crux of the algorithm lies in designing a non-greedy bidding strategy. One can see by construction the algorithm is embarrassingly parallelizable, which is favourable for GPU implementation. One can show that algorithm terminates with a valid matching and the resulting matching cost $W_U(X, Y)$ is an $\epsilon$-approximation of $w(X, Y)$. Thus, the estimate can serve as an upper bound, i.e.

$$w(X, Y) \leq W_U(X, Y) \leq (1 + \epsilon)w(X, Y), \tag{8}$$

We remark estimating Wasserstein distance $w(\mathbb{P}, \mathbb{G})$ with finite sample via primal form is only favorable in low dimensional data, such as point clouds. The error between $w(\mathbb{P}, \mathbb{G})$ and $w(X, Y)$ is $O(1/n^{1/d})$, where $d$ is data dimension (Weed & Bach, 2017). Therefore, for high dimensional data, such as images, we cannot accurately estimate wasserstein distance in primal and its upper bound with a small minibatch.

Finding a modified primal form with low sample complexity is also an open research problem (Cuturi, 2013; Genevay et al., 2018), and combining those into the proposed sandwiching objective for high dimensional data is left for future works.

### A.3 LOWER IMPLEMENTATION

The dual form of Wasserstein distance is defined as

$$w(\mathbb{P}, \mathbb{G}) = \sup_{f \in \mathcal{L}_1} \mathbb{E}_{x \sim P} f(x) - \mathbb{E}_{x \sim G} f(x), \tag{9}$$

where $\mathcal{L}_k$ is the set of $k$-Lipschitz functions whose Lipschitz constant is no larger than $k$. In practice, deep neural networks parameterized by $\phi$ with constraints $f_\phi \in \Omega_\phi$ (Arjovsky et al., 2017), result in a distance approximation

$$W_L(\mathbb{P}, \mathbb{G}) = \max_{f_\phi \in \Omega_\phi} \mathbb{E}_{x \sim P} f_\phi(x) - \mathbb{E}_{x \sim G} f_\phi(x). \tag{10}$$

If there exists $k$ such that $\Omega_f \subseteq \mathcal{L}_k$, then $W_L(\mathbb{P}, \mathbb{G})/k \leq w(\mathbb{P}, \mathbb{G}) \ \forall P, G$ is a lower bound. To enforce $\Omega_\phi \subseteq \mathcal{L}_k$, Arjovsky et al. (2017) propose a weight clipping constraint $\Omega_c$, which constrains every weight to be in $[-c, c]$ and guarantees that $\Omega_c \subseteq \mathcal{L}_k$ for some $k$. However, choosing clipping range $c$ is non-trivial in practice. Small ranges limit the capacity of networks, while large ranges result in numerical issues during the training. On the other hand, in addition to weight clipping, several constraints (regularization) have bee proposed with better empirical performance, such as gradient penalty (Gulrajani et al., 2017) and $L^2$ ball (Mroueh & Sercu, 2017). However, there is no guarantee the resulted functions are still Lipschitz or the resulted distances are lower bounds of Wasserstein distance. To take the advantage of those regularization with the Lipschitz guarantee, we propose a simple variation by combining weight clipping, which always ensures Lipschitz functions.

**Lemma 2.** *There exists $k > 0$ such that*

$$\max_{f \in \Omega_c \cap \Omega_\phi} \mathbb{E}_{x \sim P}[f_\phi(x)] - \mathbb{E}_{x \sim G}[f_\phi(x)] \leq \frac{1}{k} w(\mathbb{P}, \mathbb{G}) \tag{11}$$

Note that, if $c \to \infty$, then $\Omega_c \cap \Omega_\phi = \Omega_\phi$. Therefore, from Proposition 2, for any regularization of discriminator (Gulrajani et al., 2017; Mroueh & Sercu, 2017; Mroueh et al., 2017), we can always combine it with a weight clipping constraint $\Omega_c$ to ensure a valid lower bound estimate of Wasserstein distance and enjoy the advantage that it is numerically stable when we use large $c$ compared with original weight-clipping WGAN (Arjovsky et al., 2017).

In practice, we found combing $L^2$ ball constraint and weight-clipping leads to satisfactory performance. We also studied popular WGAN-GP (Gulrajani et al., 2017) with weight clipping to ensure Lipschitz continuity of discriminator, but we found $L^2$ ball with weight clipping is faster and more numerically stable to train.

## B   TECHNICAL PROOF

**Lemma 1.** *Suppose we have two approximators to Wasserstein distance: an upper bound $W_U$ and a lower $W_L$, such that $\forall P, G : (1 + \epsilon_1)w(\mathbb{P}, \mathbb{G}) \leq W_U(\mathbb{P}, \mathbb{G}) \leq (1 + \epsilon_2)w(\mathbb{P}, \mathbb{G})$ and $\forall P, G : (1 - \epsilon_2)w(\mathbb{P}, \mathbb{G}) \leq W_L(\mathbb{P}, \mathbb{G}) \leq (1 - \epsilon_1)w(\mathbb{P}, \mathbb{G})$ respectively, for some $\epsilon_2 > \epsilon_1 > 0$ and $\epsilon_1 > \epsilon_2/3$. Then, using the sandwiched estimator $W_s$ from (6), we can achieve tighter estimate of the Wasserstein distance than using either one estimator, i.e.*

$$\exists s : |W_s(\mathbb{P}, \mathbb{G}) - w(\mathbb{P}, \mathbb{G})| < \min\{|W_U(\mathbb{P}, \mathbb{G}) - w(\mathbb{P}, \mathbb{G})|, |W_L(\mathbb{P}, \mathbb{G}) - w(\mathbb{P}, \mathbb{G})|\} \tag{12}$$

*Proof.* We prove the claim by show that LHS is at most $\epsilon_1$, which is the lower bound for RHS.

$$
\begin{aligned}
&|W_s(\mathbb{P}, \mathbb{G}) - w(\mathbb{P}, \mathbb{G})| \\
&\quad = |(1-s)W_U(\mathbb{P}, \mathbb{G}) + sW_L(\mathbb{P}, \mathbb{G}) - w(\mathbb{P}, \mathbb{G})| \\
&\quad = |(1-s)(W_U(\mathbb{P}, \mathbb{G}) - w(\mathbb{P}, \mathbb{G})) - s(w(\mathbb{P}, \mathbb{G}) - W_L(\mathbb{P}, \mathbb{G}))| \\
&\quad \leq \max\{(1-s)\underbrace{(W_U(\mathbb{P}, \mathbb{G}) - w(\mathbb{P}, \mathbb{G}))}_{\leq \epsilon_2}, s\underbrace{(w(\mathbb{P}, \mathbb{G}) - W_L(\mathbb{P}, \mathbb{G}))}_{\leq \epsilon_2}\} \\
&\qquad - \min\{(1-s)\underbrace{(W_U(\mathbb{P}, \mathbb{G}) - w(\mathbb{P}, \mathbb{G}))}_{\geq \epsilon_1}, s\underbrace{(w(\mathbb{P}, \mathbb{G}) - W_L(\mathbb{P}, \mathbb{G}))}_{\geq \epsilon_1}\} \\
&\quad \leq \max\{(1-s), s\}\epsilon_2 - \min\{(1-s), s\}\epsilon_1
\end{aligned}
\tag{13}
$$

Without loss of generality we can assume $\lambda < 0.5$, which brings us to

$$
|W_s(\mathbb{P}, \mathbb{G}) - w(\mathbb{P}, \mathbb{G})| \leq (1-\lambda)\epsilon_2 - \lambda\epsilon_1
\tag{14}
$$

Now if we chose $\frac{\epsilon_2 - \epsilon_1}{\epsilon_2 + \epsilon_1} < \lambda < 0.5$, then $|W_s(\mathbb{P}, \mathbb{G}) - w(\mathbb{P}, \mathbb{G})| < \epsilon_1$ as desired. $\qquad\square$

**Lemma 2.** *There exists $k > 0$ such that*

$$
\max_{f \in \Omega_c \cap \Omega_\phi} \mathbb{E}_{x \sim P}[f_\phi(x)] - \mathbb{E}_{x \sim G}[f_\phi(x)] \leq \frac{1}{k}w(\mathbb{P}, \mathbb{G})
\tag{15}
$$

*Proof.* Since there exists $k$ such that $\max_{f \in \Omega_c} \mathbb{E}_{x \sim P}[f_\phi(x)] - \mathbb{E}_{x \sim G}[f_\phi(x)] \leq \frac{1}{k}w(\mathbb{P}, \mathbb{G})$, it is clear that

$$
\max_{f \in \Omega_c \cap \Omega_\phi} \mathbb{E}_{x \sim P}[f_\phi(x)] - \mathbb{E}_{x \sim G}[f_\phi(x)] \leq \max_{f \in \Omega_c} \mathbb{E}_{x \sim P}[f_\phi(x)] - \mathbb{E}_{x \sim G}[f_\phi(x)] \leq \frac{1}{k}w(\mathbb{P}, \mathbb{G}).
\tag{16}
$$

$\square$

## C  LARGER RESULTS

The larger and more hierarchical sampling discussed in Section 5.2 can be found in Figure 10. The reconstruction results on unseen classes are shown in Figure 11.

## D  ADDITIONAL STUDY

### D.1  INTERPOLATION BETWEEN ROTATIONS

It is also popular to show intra-class interpolation. In addition show simple intra-class interpolations, where the objects are almost aligned, we present an interesting study on interpolations between rotations. During the training, we only rotate data with 8 possible angles for augmentation, here we show it generalizes to other unseen rotations as shown in Figure 12.

However, if we linearly interpolate the code, the resulted change is scattered and not smooth as shown in Figure 12. Instead of using linear interpolation, We train a 2-layer MLP with limited hidden layer size to be 16, where the input is the angle, output is the corresponding latent representation of rotated object. We then generate the code for rotated planes with this trained MLP. It suggests although the transformation path of rotation on the latent space is not linear, it follows a smooth trajectory[2]. It may also suggest the *geodesic* path of the learned manifold may not be nearly linear between rotations. Finding the geodesic path with a principal method (Shao et al., 2017) and Understanding the geometry of the manifold for point cloud worth more deeper study as future work.

### D.2  CLASSIFICATION RESULTS

We evaluate the quality of the representation acquired from the learned inference network $Q$. We train the inference network $Q$ and the generator $G_x$ on the training split of ModelNet40 with data

---

[2]By the capability of 1-layer MLP.

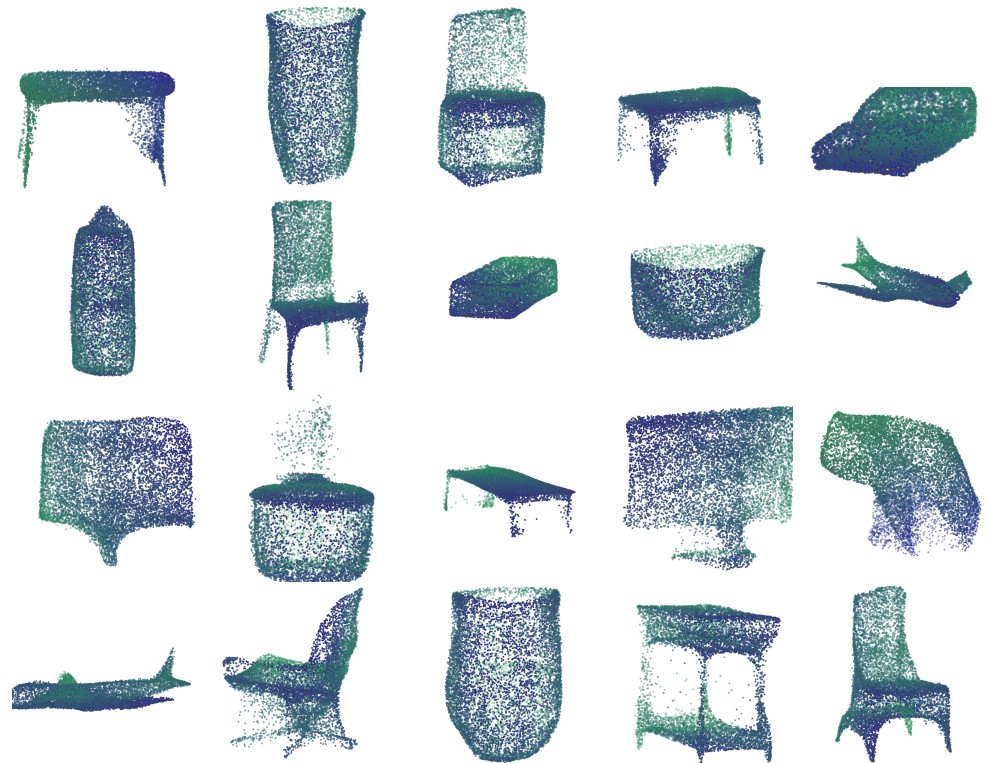

Figure 10: Randomly sampled objects and corresponding point cloud from the hierarchical sampling. Even if there are some defects, the objects are smooth, symmetric and structured. It suggests PC-GAN captures inherent patterns and learns basic building blocks of objects.

augmentation as mentioned above for learning generative models without label information. We then extract the latent representation $Q(X)$ for each point clouds and train linear SVM on the that with its label. We apply the same setting to a linear classifier on the latent code of Achlioptas et al. (2017).

We only sample 1000 as input for our inference network $Q$. Benefited by the Deep Sets architecture for the inference network, which is invariant to number of points. Therefore, we are allowed to sample different number of points as input to the trained inference network for evaluation. Because of the randomness of sampling points for extracting latent representation, we repeat the experiments 20 times and report the average accuracy and standard deviation on the testing split in Table 2. By using 1000 points, we are already better than Achlioptas et al. (2017) with 2048 points, and competitive with the supervised learning algorithm Deep Sets. We also follow the same protocol as Achlioptas et al. (2017); Wu et al. (2016) that we train on ShapeNet55 and test the accuracy on ModelNet40. Compared with existing unsupervised learning algorithms, PC-GAN has the best performance as shown in Table 3.

| Method | # points | Accuracy |
|---|---|---|
| PC-GAN | 1000 | $87.5 \pm .6\%$ |
| PC-GAN | 2048 | $87.8 \pm .2\%$ |
| AAE (Achlioptas et al., 2017) | 2048 | $85.5 \pm .3\%$ |
| Deep Sets (Zaheer et al., 2017) | 1000 | $87 \pm 1\%$ |
| Deep Sets (Zaheer et al., 2017) | 5000 | $90 \pm .3\%$ |

Table 2: Classification accuracy results.

We note that Yang et al. (2018) using additional geometry features by appending pre-calculated features with 3-dimensional coordinate as input or using more advanced grouping structure to achieve better performance. Those techniques are all applicable to PC-GAN and leave it for future works by leveraging geometry information into the proposed PC-GAN framework.

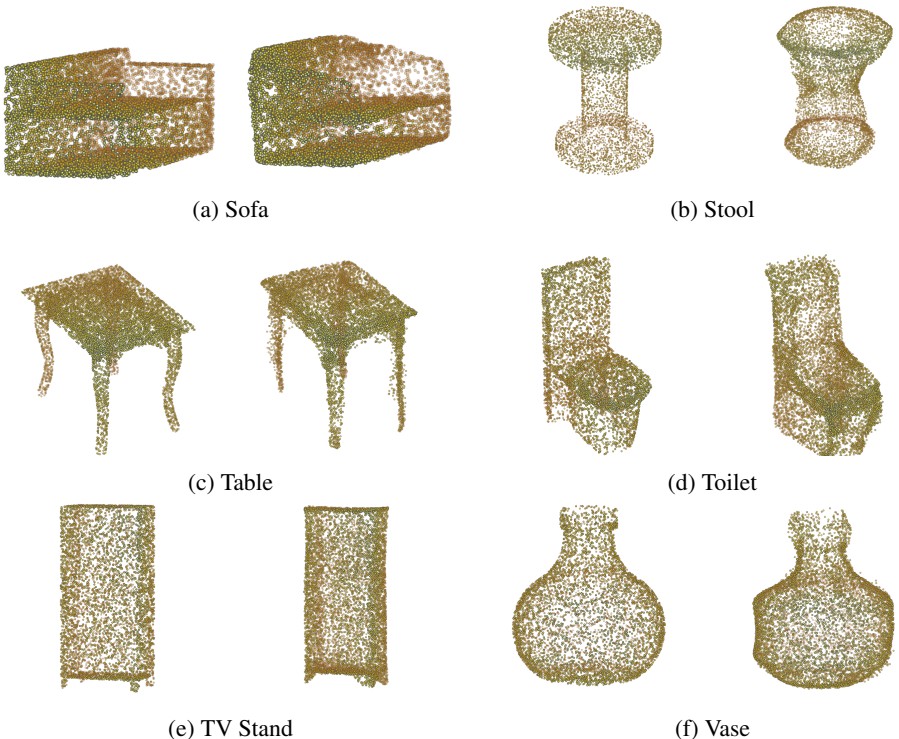

(a) Sofa             (b) Stool

(c) Table            (d) Toilet

(e) TV Stand          (f) Vase

Figure 11: The reconstructed objects from unseen categories. In each plot, LHS is true data while RHS is PC-GAN. PC-GAN generalizes well as it can match patterns and symmetries from categories seen in the past to new unseen categories.

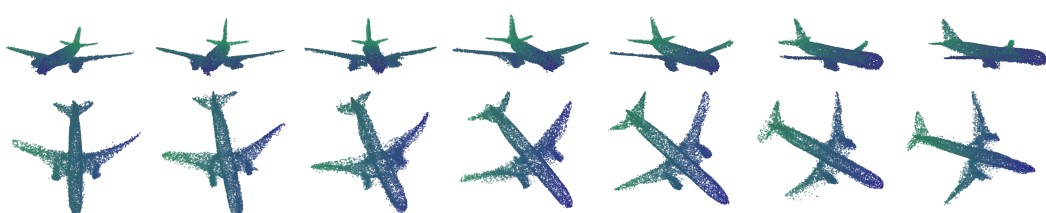

Figure 12: Interpolating between rotation of an aeroplane, using our latent space representation.

## D.3 IMAGES TO POINT CLOUD

Here we demonstrate a potential extension of the proposed PC-GAN for images to point cloud applications. After training $Q$ as described in 3 and Appendix A.1, instead of learning $G_\theta$ for hierarchical sampling, we train a regressor $R$, where the input is the different views of the point cloud $X$, and the output is $Q(X)$. In this proof of concept experiment, we use the 12 view data and the Res18 architecture in Su et al. (2015), while we change the output size to be 256. Some example results on reconstructing testing data is shown in Figure 13. A straightforward extension is using end-to-end training instead of two-staged approached adopted here. Also, after aligning objects and take representative view along with traditional ICP techniques, we can also do single view to point cloud transformation as Choy et al. (2016); Fan et al. (2017); Häne et al. (2017); Groueix et al. (2018a), which is not the main focus of this paper and we leave it for future work.

| Method | Accuracy |
|---|---|
| SPH (Kazhdan et al., 2003) | 68.2% |
| T-L Network (Girdhar et al., 2016) | 74.4% |
| LFD (Chen et al., 2003) | 75.5% |
| VConv-DAE (Sharma et al., 2016) | 75.5% |
| 3D GAN (Wu et al., 2016) | 83.3% |
| AAE (Achlioptas et al., 2017) | 84.5% |
| PC-GAN | **86.9**% |

Table 3: Classification accuracy results (Trained on ShapeNet55).

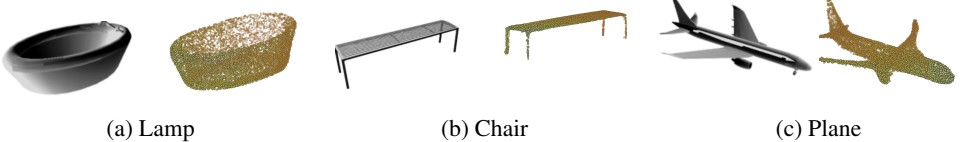

(a) Lamp                    (b) Chair                    (c) Plane

Figure 13: Image to Point Cloud

# E    DEEP SETS (PERMUTATION EQUIVARIANCE LAYERS)

We briefly review the notion of Permutation Equivariance Layers proposed by Zaheer et al. (2017) as a background required for this paper. For more details, please refer to Zaheer et al. (2017).

Zaheer et al. (2017) propose a generic framework of deep learning for set data. The building block which can be stacked to be deep neural networks is called Permutation Equivariance Layer. One Permutation Equivariance Layer example is defined as

$$f(x_i) = \sigma(x_i + \gamma\text{maxpool}(X)),$$

where $\sigma$ can be any functions (e.g. parametrized by neural networks) and $X = x_1, \ldots, x_n$ is an input set. Also, the mox pooling operation can be replaced with mean pooling. We note that PointNetQi et al. (2017a) is a special case of using Permutation Equivariance Layer by properly defining $\sigma(\cdot)$. In our experiments, we follow Zaheer et al. (2017) to set $\sigma$ to be a linear layer with output size $h$ followed by any nonlinear activation function.

# F    EXPERIMENT SETTINGS

## F.1    SYNTHETIC DATA

The batch size is fixed to be 64. We sampled 10,000 samples for training and testing.

For the inference network, we stack 3 mean Permutation Equivariance Layer (Zaheer et al., 2017), where the hidden layer size (the output of the first two layers ) is 30 and the final output size is 15. The activation function are used SoftPlus. For the generater is a 5 layer MLP, where the hidden layer size is set to be 30. The discriminator is 4 layer MLP with hidden layer size to be 30. For Achlioptas et al. (2017), we change their implementation by replcing the number of filters for encoder to be $[30, 30, 30, 30, 15]$, while the hidden layer width for decoder is 10 or 20 except for the output layer. The decoder is increased from 3 to 4 layers to have more capacity.

## F.2    MODELNET40

We follow Zaheer et al. (2017) to do pre-processing. For each object, we sampled $10,000$ points from the mesh representation and normalize it to have zero mean (for each axis) and unit (global) variance. During the training, we augment the data by uniformly rotating $0, \pi/8, \ldots, 7\pi/8$ rad on the $x$-$y$ plane. The random noise $z_2$ of PC-GAN is fixed to be 10 dimensional for all experiments.

For $Q$ of single class model, we stack 3 max Permutation Equivariance Layer with output size to be 128 for every layer. On the top of the satck, we have a 2 layer MLP with the same width and the output . The generator $G_x$ is a 4 layer MLP where the hidden layer size is 128 and output size is 3.

The discirminator is $4$ layer MLP with hidden layer size to be $128$. The random source $u$ and $z$ are set to be $64$ and $10$ dimensional and sampled from standard normal distributions.

For training whole ModelNet40 training set, we increae the width to be $256$. The generator $G_x$ is a $5$ layer MLP where the hidden layer size is $256$ and output size is $3$. The discirminator is $5$ layer MLP with hidden layer size to be $256$. For hirarchical sampling, the top generator $G_\theta$ and discriminator are all $5$-layer MLP with hidden layer size to be $256$.

For AAE, we follow every setting used in Achlioptas et al. (2017), where the latent code size is $128$ and $256$ for single class model and whole ModelNet40 models.

