# OpenReview forum: "Point Cloud GAN"
_ICLR.cc/2019/Workshop/DeepGenStruct — DeepGenStruct 2019_

### Official Review · AnonReviewer2 · 2019-04-15

**Rating:** 3
**Confidence:** 2

**Review:**

This paper proposes using GAN to generate 3D point cloud. In addition, it introduces a new "sandwiching" objective which is basically averaging the upper and lower bound of Wasserstein distance between distributions.

Although the problem this paper addresses is very important (generating 3D shapes), it has the following flaws:
1. training an inference network to model the latent variable has been done many times in the literature.
2. I am not an expert in this domain but the datasets in this paper seem to be a bit toy-ish.
3. Some more experiments need to be done on verifying the new objective.

---

### Official Review · AnonReviewer1 · 2019-04-16
**Interesting generative model for set data**

**Rating:** 4
**Confidence:** 1

**Review:**

This paper proposes a new generative model for unordered data, with a particular application to point clouds. This model includes an inference method and and a novel objective function based on sandwiching the Wasserstein distance between an upper and lower bound.  The paper includes a clear description of the problem and motivation and strong experimental results, both quantitative and qualitative.

Pros:
- Novel, reasonable model
- Clear writing
- Strong evaluation and results
- Interesting new objective based on Wasserstein bounds

Cons:
- For the practical problem of 3D shape modeling, could compare to other representations e.g. Surface Networks: http://openaccess.thecvf.com/content_cvpr_2018/papers/Kostrikov_Surface_Networks_CVPR_2018_paper.pdf

---

### Decision · Program_Chairs · 2019-04-19
**Acceptance Decision**

**Decision:**

Accept

**Comment:**

Accepted